# Theory-based electronic learning intervention to support appropriate antibiotic prescribing by nurse and pharmacist independent prescribers: an acceptability and feasibility experimental study using mixed methods

Rosemary Lim [ID],[1] Molly Courtenay,[2] Rhian Deslandes,[3] Rebecca Ferriday,[2] David Gillespie,[4] Karen Hodson,[3] Nicholas Reid,[5] Neil Thomas,[2] Angel Chater[6,7]

**Correspondence to**
Dr Rosemary Lim;
r.h.m.lim@reading.ac.uk

## ABSTRACT

**Objectives** To assess the acceptability and feasibility of using a theory-based electronic learning intervention designed to support appropriate antibiotic prescribing by nurse and pharmacist independent prescribers for patients presenting with common, acute, uncomplicated self-limiting respiratory tract infections (RTIs).

**Design** Experimental with mixed methods; preintervention and postintervention online surveys and semistructured interviews.

**Setting** Primary care settings across the UK.

**Participants** 11 nurse and 4 pharmacist prescribers.

**Intervention** A theory-based brief interactive animation electronic learning activity comprised a consultation scenario by a prescriber with an adult presenting with a common, acute, uncomplicated self-limiting RTI to support a 'no antibiotic prescribing strategy'.

**Outcome measures** Recruitment, response and attrition rates were assessed. The overall usefulness of the intervention was assessed by analysing prescribers' self-reported confidence and knowledge in treating patients with RTIs before and after undertaking the intervention, and views on the relevance of the intervention to their work. Acceptability of the intervention was assessed in semistructured interviews. The feasibility of data collection methods was assessed by recording the number of study components completed by prescribers.

**Results** 15 prescribers (maximum sample size) consented and completed all four stages of the study. Prescribers reported high to very high levels of confidence and knowledge preintervention and postintervention, with slight postintervention increases in communicating with patients and a slight reduction in building rapport. Qualitative findings supported quantitative findings; prescribers were reassured of their own practice which in turn increased their confidence and knowledge in consultations. The information in the intervention was not new to prescribers but was applicable and useful to consolidate learning and enable self-reflection.

### Strengths and limitations of this study

► To our knowledge, this study was the first to examine the acceptability and feasibility of using an electronic, interactive, animation-based learning intervention to support appropriate antibiotic prescribing by nurse and pharmacist independent prescribers for patients presenting with common, acute, uncomplicated self-limiting respiratory tract infections.

► A mixed-methods approach allowed for validation of quantitative findings; interview findings enabled a richer picture of the contextual factors affecting the feasibility and acceptability of the intervention.

► The successful recruitment of nurse and pharmacist prescribers and completion of the intervention demonstrated the acceptability and feasibility of using the intervention among nurse prescribers.

► Prescribers were an opportunistic sample, generally more experienced and may, therefore, be biased towards appropriate prescribing.

Completing the e-learning intervention was acceptable to prescribers.

**Conclusions** It was feasible to conduct the study. The intervention was acceptable and useful to prescribers. Future work will add complex clinical content in the intervention before conducting a full trial.

## INTRODUCTION

Each year, antimicrobial-resistant (AMR) infections cause approximately 700 000 deaths globally. By 2050, it has been predicted that this will rise to 10 million, combined with a cumulative cost of US$100 trillion.[1] In the European Union and the European Economic Area, the figure is an estimated

33 110 deaths, 875 000 disability-adjusted life-years[2] and €1.5 billion in extra healthcare costs.[3] The inappropriate use of antimicrobials in humans is one of the leading drivers for the growth of AMR[4] and strategies to support appropriate antibiotic use are important. A key global strategy is careful stewardship of antibiotics.[1] Antimicrobial stewardship (AMS), comprises 'a collection of coordinated interprofessional focused strategies to optimise antibiotic use by ensuring that every patient receives an antibiotic only when it is clinically indicated and then receives the appropriate antibiotic, at the right dose, duration and route of administration'.[5]

Common, acute, uncomplicated self-limiting respiratory tract infections (RTIs) usually resolve spontaneously, with antibiotics in most cases, unlikely to offer clinical benefit.[6] These common conditions include acute otitis media (no otorrhoea, ie, discharge following an eardrum perforation and not a child under 2 years old with otitis media in both ears),[7] acute sore throat/acute pharyngitis/acute tonsillitis (FeverPAIN score 0 or 1 or Centor scores 0, 1 and 2),[8] acute sinusitis (symptoms lasting 10 days or less)[9] and acute cough (associated with an upper RTI and acute bronchitis).[10] Despite this, more than 60% of all prescriptions issued in the UK primary care are for RTIs.[11 12] As well as contributing to the spread of resistance, their unnecessary use also puts patients at risk of side effects.[13] A global priority that has been recognised for some time is the need to conserve antibiotic sensitivity by managing RTIs without recourse to antibiotics and healthcare professionals who routinely prescribe antibiotics are a key target for interventions.[13–16]

Multifaceted interventions that address barriers to change in specific healthcare settings involving active education strategies, feedback on antibiotic prescribing behaviour and seeking to improve prescribing for all as opposed to specific respiratory infections, tend towards greater effectiveness in medical prescribers.[17 18] Comparable reductions in the utilisation of antibiotics by medical prescribers have been demonstrated using these strategies via electronic learning.[19]

Appropriately qualified nurses and pharmacists in the UK can prescribe medicines independently and around 34 000 nurses and 9000 pharmacists have independent prescribing capability.[20] Nurse and pharmacist prescribers frequently manage patients with RTIs and prescribe around 8% of all primary care antibiotics dispensed in England.[21] As compared with medical prescribers, a broader range of factors influences the prescribing behaviour of these professionals.[22] Factors include diagnostic uncertainty and the patient's clinical condition,[22–25] the expectations of patients for an antibiotic,[22 23 25] relationships with other prescribers and knowledge of current guidelines.[22 23 26] Interventions are available to support the various AMS activities that healthcare professionals are involved in[27 28]; however, no interventions exist to specifically support appropriate antibiotic prescribing behaviour by nurse and pharmacist independent prescribers. We developed a theory-based electronic learning intervention that aimed to support a 'no antibiotic prescribing strategy' by nurse and pharmacist independent prescribers for common, acute, uncomplicated self-limiting RTIs to reflect national (The National Institute for Health and Care Excellence) prescribing guidance. The aim of this study was to assess the acceptability and feasibility of implementing this e-learning intervention in UK general practice.

## METHOD

### Design

A mixed-methods study design was used: participants completed an online preintervention and postintervention survey, participated in the intervention and a semistructured interview with a researcher.

### Participant recruitment

The inclusion criteria for participants were nurse and pharmacist non-medical independent prescribers (hereafter described as prescribers) who managed patients with RTIs in a primary care setting in the UK. Recruitment took place from November to December 2018 via several routes until a maximum of 15 nurse and pharmacist prescribers in total had been recruited; a sample size expected to enable qualitative data saturation[20]: (1) MC emailed prescribers (4 pharmacists and 17 nurses) who consented to be contacted after taking part in previous research led by MC[22]; (2) KH and RL approached key contacts within their existing prescriber networks who sent out information about the study via email (approximately 195 pharmacists and nurses combined). MC or RL emailed the participant information sheet and consent form to interested prescribers and encouraged them to ask any questions they may have about the study prior to making an informed choice about participating in the study. Participants gave written informed consent prior to taking part in the study.

### Intervention

The theory-based intervention comprised a 5-minute, interactive, animated scenario of a consultation by a prescriber with an adult presenting with a common, acute, uncomplicated self-limiting RTI. The development of the intervention is published elsewhere[20] and draws from previous work with nurse prescribers[29 30] and pharmacists.[31 32] The behaviour change wheel (BCW)[33] three-stage, eight-step approach was used to design the intervention content taking into consideration Capability, Opportunity and Motivation-Behaviour (COM-B) and Theoretical Domains Framework (TDF). The COM-B was used to create a behavioural diagnosis that is, what needs to change; and the TDF identified relevant psychosocial drivers of the behaviour. The intervention functions, 'education', 'training' and 'modelling' were identified as the most appropriate for changing AMR prescribing behaviour, using the COM-B mapping from the BCW. The development of the e-learning activity was underpinned by 'gamification' as a teaching approach,

specifically using a 'spot the difference' format animation. There is increasing evidence of improved knowledge among healthcare professionals on sepsis management[34] and antibiotic use,[35] following usage of online games despite previous inconclusive evidence of the utility of educational games for health professionals.[36]

Prescribers were sent a weblink to the intervention that was accessible on any internet-enabled device. They were asked to engage with the e-learning activity by watching and answering questions posed within the activity. The intervention comprised two scenarios with the first depicting a consultation by a prescriber to reach a no antibiotic prescribing decision whereby patient-centred approaches,[37] such as holistic care, individualised care, respectful care and empowering care, to reach a prescribing decision were absent. The overall patient experience and satisfaction were poor; the patient was not confident in the treatment decision made in the consultation. This is followed by a second scenario where a prescriber used a patient-centred motivational interviewing style[38] to reach a no antibiotic prescribing decision in the consultation. In this scenario, the patient left the consultation satisfied and confident with the treatment received. To facilitate active learning, prescribers were explicitly asked to 'spot the difference' between the two scenarios by answering a range of open and closed questions that were incorporated in this second scenario. These questions focused on the different approaches the second prescriber undertook to consult with the patient.

### Measures and data collection
The study was conducted in four stages in the following order: (1) preintervention online questionnaire, (2) e-learning intervention, (3) postintervention online questionnaire and (4) semistructured telephone interviews. The overall usefulness of the intervention was assessed by analysing prescribers' self-reported confidence and knowledge in treating patients with RTIs before and after undertaking the intervention, and views on the relevance of the intervention to their work. Acceptability of the intervention was assessed in semistructured interviews. The feasibility of data collection methods was assessed by recording the number of study components completed by prescribers. Responses to open and closed questions in the e-learning intervention were not collated and analysed because the aim of the study was to assess the acceptability and feasibility of implementing the intervention.

### Pre–post intervention online questionnaires
Prescribers were sent a link and completed an online questionnaire before and immediately after completing the intervention aimed to assess their perceptions of the impact of the intervention. The questionnaires were developed based on findings from our previous work.[20] The preintervention questionnaire assessed prescribers' knowledge and confidence prescribing antibiotics for patients presenting with RTIs. Prescribers were asked to rate their responses on six items using a 5-point Likert

scale (strongly disagree to strongly agree): (1) gain information on patient expectations, (2) support patients, (3) build rapport, (4) communicate effectively, (5) see and examine different viewpoints, and (6) ensure that patients both understand and are happy with the prescribing decision. The postintervention questionnaire contained the same questions as the preintervention questionnaire, and six additional questions about the usefulness of the intervention: (1) whether the information was known to participants, (2) its applicability to practice, (3) whether the intervention would be useful to them as prescribers, (4) whether it makes them feel more comfortable when speaking with patients with RTIs, (5) if it encourages participants to consider how they would apply the information to practice and (6) think differently. In both questionnaires, demographic details were also collected: type of prescriber (ie, nurse or pharmacist), length of time qualified as a prescriber, time in the current post, clinical setting and length of consultation time.

### Semistructured interviews
Following the completion of the intervention and the postintervention questionnaire, prescribers were invited to take part in a semistructured telephone interview to understand prescribers' experiences of using the intervention. The interview took place within 1–2 weeks postintervention to ensure the retention of information. Interviews (see box 1 for the interview schedule) explored each component of the COM-B model, the hub of the BCW[31] to create a behavioural diagnosis to understand capability (eg, knowledge and skills), opportunity (eg, social influence and environment) and motivation (eg, beliefs about consequences and emotional responses). Demographic details such as job title, practice setting, years qualified as a prescriber and approximate frequency of RTI consultations and antibiotic prescribing were also collected in the interview. All interviews were digitally recorded and then transcribed verbatim with identifying information removed.

### Data analysis
Quantitative data were collected prior to the qualitative interviews and both datasets were analysed separately but within the same time frame. Quantitative findings informed the qualitative interviews and the qualitative findings explained in more detail, the quantitative findings. An iterative process of comparing both the quantitative and qualitative datasets allowed for data triangulation and confirmed the accuracy of findings in the interviews, providing further insight into the phenomenon under study.[39 40]

### Pre–post intervention online questionnaires
Data were analysed using SPSS V.25.[41] Descriptive statistics were used to characterise prescribers (frequencies and percentages), prescribers' knowledge and confidence in prescribing antibiotics for patients presenting with RTIs

## Box 1 Interview schedule

**Capability**
Examples of prompts:
► Did the learning resource increase your knowledge and skills about prescribing antibiotics for respiratory tract infections (RTIs)?
► (If yes) Can you tell me in what ways?
► (If no) Can you tell me why this was the case for you?
► Did it increase your awareness of potential solutions to any difficulties you may have experienced managing the consultations of patients with RTIs?
► (If yes) Can you tell me in what ways?
► (If no) Can you tell me why this was the case for you?
► How do you think your practice will change based on the training resource?

**Opportunity (relating to the resource and also behaviour to be changed)**
Examples of prompts:
► How do you think this resource would change the norms of practice? Or the practice of others. How will this fit in to current practice? (ie, in terms of time)
► How did the resource address any gaps in your prescribing practice?
► Is the resource an acceptable/feasible delivery method that can be integrated into daily practice?

**Motivation**
Examples of prompts:
► In what way does the resource provide a means by which you can reflect on and develop your practice?
► How do you perceive that this resource could improve prescribing practice generally?
► How did it make you feel about their current and future practice?

**Other questions**
► What stood out for you the most, in the learning resource? For example, was it the poster; was it that she came to greet the patient, etc.
► How can the learning resource be improved? Was there anything that you would change for future training?

### Table 1 Demographic data of participants (n=15)

| | |
|---|---|
| Type of non-medical prescriber | |
| Nurse | 11 (73.3%) |
| Pharmacist | 4 (27.7%) |
| Time qualified as the prescriber | |
| 13 months to 2 years | 3 (20.0%)* |
| 3–5 years | 5 (33.3%) |
| 6–10 years | 2 (13.3%) |
| >10 years | 5 (33.3%) |
| Time in post | |
| 13 months to 2 years | 4 (26.7%) |
| 3–5 years | 9 (60.0%) |
| 6–10 years | 0 (0.0%) |
| >10 years | 2 (13.3%) |
| Type of clinical setting and length of patient appointments | |
| Community care (45–90 min) | 2 (13.3%) |
| General practice (10–15 min) | 10 (66.7%)† |
| Out of hours (10–20 min) | 1 (6.7%) |
| General practice and out of hours (10–30 min) | 1 (6.7%) |
| Urgent care (15 min) | 1 (6.7%) |

*Two of the three participants were pharmacists.
†All pharmacists worked in general practice and had 15 min of appointment times.

(mean, SD) and their views on the usefulness of the intervention (mean, SD).

### Semistructured interviews

Interviews were analysed using inductive thematic analysis.[42] Coding and categorising of data were conducted by RL using NVivo V.10.[43] Themes were then identified reviewed with a second researcher (MC) and any differences in interpretation were resolved through discussion to increase the trustworthiness of research data.[44 45]

### Patient and public involvement

Patients and the public were not involved in the development of the research question, study design, recruitment and conduct of the study.

## RESULTS

A total of 15 of 216 prescribers approached (7%) responded to the study invitation, consented and completed all four stages of the study between October and December 2018.

The maximum number of participants targeted for the study was reached. Table 1 shows the demographic data of prescribers. Most prescribers worked in general practice (n=10, 67%) and had been qualified for at least 2 years (n=12, 80%). Except for prescribers in the community care setting, most prescribers worked within 10–30 min of patient appointment times. Interviews with prescribers lasted between 7 and 28 min (mean=16 min).

### Impact of intervention on prescribing practice

Table 2 shows prescribers' scores relating to their confidence in managing patients presenting with RTIs preintervention and postintervention. High to very high levels of confidence were reported for all statements both preintervention and postintervention. There was an increase in confidence levels in statements 1.2 (supporting patients understand health information given), 1.4 (skills to communicate with patients) and 1.5 (skills to help patients see and examine different viewpoints). Participants scored very highly for statement 1.3 (building rapport with patients) both preintervention and postintervention but there was a slight reduction in confidence postintervention. Prescribers' confidence in their own ability to gain health-related information and that patients understand and are happy with their prescribing decision stayed the same.

Interviews with prescribers supported quantitative findings and revealed that watching and learning 'good

**Table 2** Confidence in the treatment management of patients with RTI preintervention and postintervention

| Statement | Preintervention | | Postintervention | | Mean change |
|---|---|---|---|---|---|
| | Min, max | Mean (SD) | Min, max | Mean (SD) | |
| 1.1 Confidence in gaining health-related information | 3, 5 | 4.20 (0.68) | 3, 5 | 4.20 (0.56) | 0 |
| 1.2 Confidence in supporting patients understand health information given | 4, 4 | 4.00 (0.00) | 4, 5 | 4.27 (0.46) | 0.27 |
| 1.3 Confidence in building rapport with patients | 4, 5 | 4.53 (0.52) | 4, 5 | 4.47 (0.52) | −0.06 |
| 1.4 Confidence in skills to communicate with patients | 4, 5 | 4.53 (0.52) | 3, 5 | 4.60 (0.63) | 0.07 |
| 1.5 Confidence in skills to help patients see and examine different viewpoints | 3, 5 | 3.87 (0.74) | 3, 5 | 4.13 (0.64) | 0.26 |
| 1.6 Confidence that patients understand and happy with prescribing decision | 2, 5 | 4.07 (0.80) | 3, 5 | 4.07 (0.46) | 0 |

RTI, respiratory tract infection.

practice', as shown in the intervention, reassured them of their own practice. It appeared to increase their confidence in their current practice and in refusing to prescribe antibiotics, when appropriate.

> So, when I watched it, I was thinking, well, yeah, I do, it reinforced what I was doing was right… and I just, you know, keep doing that. (Nurse 10)

> It's given me more confidence in my approach really, to managing those patients with the respiratory tract infections. (Pharmacist 4)

### Usefulness of intervention

Table 3 shows the prescribers' level of agreement with statements relating to the usefulness of the intervention. Responses were recorded on a 5-point Likert scale: 1=strongly disagree, 2=somewhat disagree, 3=neither agree nor disagree, 4=somewhat strongly agree and 5=strongly agree. Although prescribers disagreed that the information in the intervention was new to them, they agreed that the intervention was applicable and would be useful to them as prescribers.

Qualitative data were consistent with the questionnaire data; the majority of prescribers did not think that they had acquired new knowledge or skills from the e-learning intervention because they were largely an experienced group of practitioners. For most prescribers, however, the intervention was useful in helping them refresh their memories, consolidate learning and prompted self-reflection on prescribing for patients presenting with RTIs.

> I've been doing it quite a long time now and, you know, especially in my respiratory clinic, so I suppose, some of it was just reinforcing what I already sort of knew, but then that's good as well. It's still teaching, isn't it? It's still learning? (Nurse 6)

> Yeah, I think it's about, you know, I like to do snippets of education and I sell it as… and this is very much this. It's the cement between the bricks. Your day's bricks and this is just a little filler that builds it all together. (Nurse 3)

> I think to… to quite a big extent, I've probably been prescribing more than I needed to…I often deal with the very frail and elderly. I often deal with the very frail and elderly. And urine infections the whole time… in the past, I have really thought, oh, crikey, they're old, they must have antibiotics. But actually, that's not necessarily a good thing. (Nurse 11)

Some learnt other methods of communicating with patients, for example, the use of additional information

**Table 3** Usefulness of intervention from the postintervention questionnaire

| Statement | Minimum | Maximum | Mean | SD |
|---|---|---|---|---|
| 2.1 The information was mostly new to me | 1 | 2 | 1.33 | 0.49 |
| 2.2 The intervention was applicable to my practice | 2 | 5 | 4.60 | 0.83 |
| 2.3 The intervention will be useful to me as a prescriber | 3 | 5 | 4.20 | 0.78 |
| 2.4 The intervention has made me feel more comfortable speaking with patients with RTIs | 2 | 4 | 3.13 | 0.83 |
| 2.5 The intervention has encouraged me to consider how I would apply the information in my practice | 2 | 5 | 3.87 | 0.99 |
| 2.6 The intervention has encouraged me to think differently | 1 | 4 | 3.07 | 1.10 |

RTI, respiratory tract infection.

such as posters and leaflets and scheduling follow-up appointments after the consultation as a 'safety net'.

> I would never really have thought to… to contact the patients… I'll think about doing that in the future. And who… who are worried, um, about not prescribing antibiotics for them, um, I think that… that would ease the pressure from it. (Pharmacist 4)

Despite claiming not learning new knowledge, some prescribers said they intended to make changes to their practice specifically around involving the patient more in decision-making and directing patients to relevant information resources. A few prescribers claimed to have already changed their practice, for example, in considering different approaches to arrive at treatment decisions and using different ways of communicating with patients.

> I'll be sort of more vigilant in making sure that, you know, the patient understands why, I've, you know, decided to go with a certain treatment and sort of try and get the patient to engage and agree with that decision…So they're involved within the decision and I think that kind of gives them a better understanding. (Pharmacist 1)

> …having a simple resource that I can always apply to my consultations…was actually really helpful for that and I've used it in that context a number of times with different patients. With palpable different outcomes really, I think, you know. I have, you know, embarked on the education, rather than prescription approach, in… in consequence to doing it, so it has been good for that. (Nurse 4)

> I began to think much more clearly in the terms of, give me a reason to give them antibiotics. So I think that was a little bit of a theme change for me. Because I'd always thought, oh, I don't want to upset… well, I don't like upsetting people. I don't. I like people to go away feeling as if I've listened to them… And they maybe have not got what they came in to look for, but they're satisfied. (Nurse 11)

### Acceptability of the intervention
#### Design and content of the intervention
Prescribers said they liked the use of e-learning because there was the flexibility of where and when learning can take place. The use of 'cartoon' characters and scenes generated mixed views. The scenario-based learning approach was well-received; the scenario was realistic and memorable.

> And that will… stay with me. You know, when I think about it, that's the vision that I have, is this person typing really fast. It's just… it was just funny. I just thought, God, that's all of us. We're just typing really fast, thinking come on! Hurry up, hurry up! (Nurse 1)

I liked the scenario around it as that seemed real. And it's… it's reassuring actually those conversations that were had in the video, which we all know we all have…It kind of just said, you know, we do have these difficult things. You know, patients sometimes come in, they want antibiotics and they… you know, they perceive the need for them, but it was nice to see that we can say, you don't need them, it's okay. (Nurse 3)

Prescribers said the messages presented in the e-learning intervention were appropriate, relevant, easy to understand and consistent with their own previous learning and practice. Some messages that stood out included managing patient expectations by using posters in the waiting room and providing self-care advice.

> I know and I know that, um, my GP colleagues also, we all struggle with the patient expectation of antibiotics, so I liked very much the idea of managing expectations of having… having the posters in the waiting room, of having the literature available. (Nurse 2)

Some prescribers highlighted the use of effective communication and observational skills in the intervention to be valuable aspects of the intervention.

> …you know when she tells the, er, patient the second time round why she's not giving a… why she doesn't need antibiotics… I think that that part really stood out for me. Because I think it was explained clearly, it was precise, um, it wasn't too long, it wasn't too short. And it was just… just right and I think that the level of language used. What was… I really liked it. I think it was spot on really. (Pharmacist 1)

#### Completing the intervention
All the prescribers said it was quick (approximately 5–20 min) to complete the intervention. Around half of the prescribers said they completed the intervention at work for example during their lunch break, in between appointments, whereas the rest completed the intervention in their own time. Despite this, prescribers agreed that the intervention could be completed during work time because it was a short learning session.

### Suggestions for improving the e-learning intervention
#### Perceived and potential demographic of learners
Most prescribers said that the content in the e-learning intervention was pitched at a lower level of experience and suggested additional groups of people who may benefit from it such as General Practitioners (GP), GP trainees, new medical and non-medical prescribers, undergraduate healthcare students and the public.

> I thought at times that the level was a little bit lower. I thought the level being pitched was a little bit low and I don't know if that was intentional. (Nurse 3)

> I suppose what the content of the… the little video was, was fairly sort of low level, in terms of antimicrobial

prescribing. I suppose it's the sort of thing that I would expect maybe most people prescribing in that environment to know. (Nurse 9)

## Access

A few prescribers reported experiencing problems accessing the intervention at work due to the National Health Service (NHS) firewall. They circumvented the problem by using their own personal device.

## Overall presentation

A few prescribers suggested using a more diverse set of characters for example, gender, race, age and a more realistic layout of a consulting room.

## Content

Other suggestions included adding learning outcomes to set learners' expectations, providing additional information such as an estimated time to complete the intervention, context to the scenario and patient history, more information about aspects shown in the learning intervention, for example, relevance and information about the poster, physical examinations, highlight key messages and including questions on self-reflection within the intervention. There was also a suggestion to include subtitles to the scene. Some prescribers suggested adding scenarios with different challenges, for example, a difficult patient, different groups of patients and patient revisits. To accommodate different learner requirements, some prescribers suggested including different levels of clinical content such as the most up-to-date clinical guidelines and additional reading materials.

## DISCUSSION

Completing the e-learning intervention was acceptable to prescribers. It was also feasible to collect study data in four separate stages. Overall, prescribers reported positive views about the usefulness of the intervention. Prescribers reported high levels of confidence in managing patients with RTIs both preintervention and postintervention. Although the recruitment of prescribers to the study was successful, the recruitment of pharmacist prescribers was challenging. Nationally, there are fewer pharmacist prescribers compared with nurse prescribers[21] and further consideration to sample size will be needed to inform the next stage of this study.

The intervention component of the study was short, around 5 min, and all prescribers were able to complete the intervention at a time and place that suited them. Although some prescribers chose to complete the intervention in their personal time, all agreed that it was feasible to complete the intervention as part of their day-to-day work suggesting the acceptability of using the intervention as part of their training. There was overall support for using an e-learning mode of delivery and this was consistent with the suggestion made by a similar group of prescribers[15] however, there were reported issues with

accessing the intervention on NHS computers. The next iteration of the intervention will need to consider wider access issues if the learning were to take place as part of routine training in the future.

Prescribers reported high to very high levels of confidence for all statements both preintervention and postintervention and the finding was consistent with the qualitative dataset. The findings could be explained by the demographics of the study population—a group of prescribers who were largely very experienced in their role; nurse and pharmacist prescribers have often undertaken postgraduate specialist training and although many nurse prescribers have undertaken specialist qualifications, they have not all done so and only allowed to prescribe within their area of competence.[46–48] Although there were small increases in confidence levels related to skills relating to communicating with patients and a slight reduction in building rapport with patients, it was not possible to make generalisations due to the small sample size. Prescribers were also an opportunistic sample, were generally more experienced and may therefore be biased towards appropriate prescribing. Although interviews took place as soon as possible after prescribers had completed the intervention and postintervention questionnaire, these typically took place after 2 weeks and this time lag could affect recall of their experiences of undertaking the intervention.

It is however worth noting that prescribers reported that the intervention enabled self-reflection and the slight reduction in confidence in building rapport with patients could be the result of prescribers' reflection of their practice. Regarding the usefulness of the intervention, prescribers scored low on aspects relating to learning new information in the intervention. But, prescribers generally considered the intervention to be useful because it provided the opportunity to remind themselves on the topic, enabled self-reflection and change practice. Reflecting on one's professional practice is a key requirement for pharmacist and nurse revalidation[49 50] and is an interesting outcome that was not expected or measured in this study. The brief and easily accessible intervention used in this study is similar to the learning that healthcare staff are increasingly accessing for example, 'Espresso' sessions held at workplaces, suggesting the potential for widespread adoption.

Although the development of the intervention was directly informed by research conducted with prescribers with similar demographic characteristics as those in this study and underpinned by psychological and pedagogic theories, there were aspects to the intervention that requires further refinement specifically the level of difficulty of the clinical content. Intervention development, testing and evaluation is an iterative process and should be based on a user-centred approach.[51 52] The next iteration of the e-learning intervention will focus on the addition of complex clinical content, possibly multiple activities and testing focused on the appropriateness of the clinical content.

## Strengths and limitations

To our knowledge, this study was the first to examine the acceptability and feasibility of using an electronic, interactive, animation-based learning intervention to support appropriate antibiotic prescribing by nurse and pharmacist independent prescribers for patients presenting with common, acute, uncomplicated self-limiting RTIs. A mixed-methods approach allowed for validation of quantitative findings; interview findings enabled a richer picture of the contextual factors affecting the feasibility and acceptability of the intervention. The successful recruitment of nurse and pharmacist prescribers and completion of the intervention demonstrated the acceptability and feasibility of using the intervention among nurse prescribers.

A small sample was recruited to determine the acceptability and feasibility of using the intervention. Therefore, findings will need to be interpreted with caution. Prescribers were also an opportunistic sample, were generally more experienced and may therefore be biased towards appropriate prescribing. Although interviews took place as soon as possible after prescribers had completed the intervention and postintervention questionnaire, these typically took place after 2 weeks and this time lag could affect recall of their experiences of undertaking the intervention.

## CONCLUSIONS

The study showed that it was feasible to conduct the study and the intervention was acceptable and useful to prescribers. Suggestions to improve the usefulness of the intervention focused on the clinical content rather than its delivery. Future work will consider the addition of complex clinical content in the intervention before considering a full trial.

**Author affiliations**
[1]Reading School of Pharmacy, University of Reading, Reading, UK
[2]School of Healthcare Sciences, Cardiff University, Cardiff, UK
[3]Cardiff School of Pharmacy and Pharmaceutical Sciences, Cardiff, UK
[4]Centre for Trials Research, Cardiff University, Cardiff, UK
[5]Public Health Wales, Cardiff, UK
[6]Centre for Health, Wellbeing and Behaviour Change, University of Bedfordshire, Luton, UK
[7]University College London School of Pharmacy, London, UK

**Acknowledgements** The authors would like to thank all the prescribers who took part in the study, and network contacts who helped to disseminate information about the study.

**Contributors** MC made a substantial contribution to the conception and design of the work, interpretation of data and drafting of the work. RL made a substantial contribution to the design of the work, the acquisition, analysis and interpretation of data and drafting of the work. AC and DG made a substantial contribution to the design of the work and interpretation of data. KH made a substantial contribution to the conception of the work, participant recruitment and drafting of the work. RD, RF, NR and NT made a substantial contribution to the conception of the work and drafting of the work. All authors approved the final version to be published and agree to be accountable for all aspects of the work in ensuring that questions related to the accuracy or integrity of any part of the work are appropriately investigated and resolved.

**Funding** This work was supported by the Economic and Social Research Council (ESRC) Impact Acceleration Account (IAA).

**Competing interests** None declared.

**Patient and public involvement** Patients and/or the public were not involved in the design, or conduct, or reporting, or dissemination plans of this research.

**Patient consent for publication** Not required.

**Ethics approval** Ethical approval for the study has been provided by the School of Healthcare Sciences Research Governance and Ethics Committee, Cardiff University, UK (reference: 427REC). All participants gave written informed consent before participating in the study.

**Provenance and peer review** Not commissioned; externally peer reviewed.

**Data availability statement** All data relevant to the study are included in the article or uploaded as supplementary information. Data sharing statement: No additional data available; that is, the dataset supporting the conclusions of this article is included within the article.

**ORCID iD**
Rosemary Lim http://orcid.org/0000-0003-1705-1480

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
