## [Reviewer comments · BMJ Open]

ARTICLE DETAILS

TITLE (PROVISIONAL)	A theory-based electronic learning intervention to support appropriate antibiotic prescribing by nurse and pharmacist independent prescribers: an acceptability and feasibility experimental study using mixed methods.
AUTHORS	Lim, Rosemary; Courtenay, Molly; Deslandes, Rhian; Ferriday, Rebecca; Gillespie, David; Hodson, Karen; Reid, Nicholas; Thomas, Neil; Chater, Angel

VERSION 1 – REVIEW

REVIEWER	Susanne Kaae Dep. of Pharmacy, University of Copenhagen
REVIEW RETURNED	25-Feb-2020

GENERAL COMMENTS	Overall comments: Overall the topic is important and relevant: appropriate antibiotic prescribing. The paper has a clear structure and is in general easy to follow. However, if the intervention is not described and discussed more in-depth including the theoretical and pedagogic rationales behind it, I think the paper remains superficial and challenging for the reader to achieve some specific learning with regard to how to design similar interventions in the future. Specific comments: Abstract: Intervention: is it possible to describe a bit more in detail what is the purpose of the intervention? Introduction: Is it possible to define what is meant by ‘uncomplicated self-limiting respiratory tract infection’ so all readers are aware of this when reading the article? Could you please specify the aim of your study/ intervention – what exactly did you aim for with your specific intervention and why (seems very much to do with the interaction with the patient)? Method: Did you apply any criteria when selecting participants or was the process purely convenience based? Please describe the theories on which you based your intervention. What were the reflections behind the specific content and design of the intervention – and how did this link to the theories? (also in terms of teaching and learning theory?) Please also elaborate how participants interacted with the illustrated consultations – perhaps providing a few examples. Which were the different study components that participants had to complete and which you measured?
---

	Please elaborate how you compared the quantitative and qualitative data during the analysis. Discussion Could the results be further discussed in relation to the design of the intervention (both content and pedagogy) – the discussion is right now summing up the results rather than providing the reader with some insights into how to create such interventions, what are the do’s and don’ts? It says the next stage is to evaluate the clinical content – how is that different from what was evaluated in this study (knowledge, suggestions for improvements, etc.)? and why is it not incorporated in this study? It would also be interesting to discuss how much you can achieve by providing short, flexible, feasible interventions as compared to an intervention being a bit longer? Hence, how to balance time with learning outcome? Please apart from discussing generalization challenges due to a small sample size and due to participants in general being experienced, could you also reflect how the way in which you recruited participant might have influenced the results – hence, was there a selection bias which might have played a role?
--	---

REVIEWER	Stan Deresinski Stanford University United States
REVIEW RETURNED	27-Feb-2020

GENERAL COMMENTS	This paper describes a pilot study evaluating the feasibility acceptability to learners of a 5 minute interactive animation e-learning activity focusing on a simulated adult patient with a self-limiting upper respiratory tract infection. The learners were a highly select group consisting of 11 nurses and 4 pharmacists representing a response rate of only 7% of the prescribers contacted, at least some of whom were contacted because they had participated in previous learning experiences. The basic findings were that the the e-learning experience was feasible and that the evaluators indicated that it was acceptable. While the activity was associated with a self-reported improved confidence in dealing with patients represented by the scenario by enforcing their preexisting knowledge and experience, the participants, 80% had been qualified for >2 years, indicated that the activity had “aimed too low”. Thus, a major lesson of this experience is the difficulty of gauging the appropriate level of complexity in an activity such as this – something which may require iterative trials at various levels and the development of multiple e-learning activities geared to various levels of training and experience.
---

VERSION 1 – AUTHOR RESPONSE

Reviewer 1	
Overall comments: Overall the topic is important and relevant: appropriate antibiotic prescribing. The paper has a clear structure and is in general easy to follow. However, if the intervention is not described and discussed more in-depth including the theoretical and pedagogic rationales behind it, I think the paper remains superficial and challenging for the reader to achieve some specific learning with regard to how to design similar interventions in the future.	We agree with the reviewer’s view on the need to include the theoretical and pedagogic rationales that underpinned our intervention design and evaluation. We have now added further details in the method section – please also see our response to a similar comment under ‘methods’ in this table.
Specific comments: Abstract: Intervention: is it possible to describe a bit more in detail what is the purpose of the intervention?	The purpose/the target behaviour of the intervention has now been added: Intervention: A theory-based brief interactive animation electronic learning activity comprised of a consultation scenario by a prescriber with an adult presenting with a common, acute, uncomplicated self-limiting RTI to support a ‘no antibiotic prescribing strategy.’
Introduction: Is it possible to define what is meant by ‘uncomplicated self-limiting respiratory tract infection’ so all readers are aware of this when reading the article?	We have now specified the conditions in the text, which now reads as follow: Common, acute, uncomplicated self-limiting respiratory tract infections (RTIs) usually resolve spontaneously, with antibiotics in most cases, unlikely to offer clinical benefit [6]. These common conditions include acute otitis media (no otorrhoea i.e. discharge following an ear drum perforation and not a child under 2 years old with otitis media in both ears) [7], acute sore throat/acute pharyngitis/acute tonsillitis (FeverPAIN score 0 or 1 or Centor score 0, 1, 2) [8], acute sinusitis (symptoms lasting 10 days or less) [9] and acute cough (associated with an upper respiratory tract infection and acute bronchitis) [10].

Introduction: Could you please specify the aim of your study/ intervention – what exactly did you aim for with your specific intervention and why (seems very much to do with the interaction with the patient)?	We have amended the text considering the comment which now reads: We developed a theory-based electronic learning intervention that aimed to support a ‘no antibiotic prescribing strategy’ by nurse and pharmacist independent prescribers for common, acute, uncomplicated self-limiting RTIs to reflect national (NICE) prescribing guidance. The aim of this study was to assess the acceptability and feasibility of implementing this e-learning intervention in UK general practice.
Method: Did you apply any criteria when selecting participants or was the process purely convenience based?	We have made clearer our inclusion criteria as follows: The inclusion criteria for participants were nurse and pharmacist non-medical independent prescribers (hereafter described as prescribers) who managed patients with RTIs in a primary care setting in the UK.
Method: Please describe the theories on which you based your intervention. What were the reflections behind the specific content and design of the intervention – and how did this link to the theories? (also in terms of teaching and learning theory?)	The development of the intervention is detailed in our published protocol paper (https://bmjopen.bmj.com/content/9/8/e028326). We have also included a brief description of the underpinning theory to the development of our intervention in the manuscript. The intervention was designed using the Behaviour Change Wheel (BCW) [33]. The BCW three-stage, eight-step approach was used, which included the development of the intervention content taking into consideration Capability, Opportunity and Motivation-Behaviour (COM-B) and Theoretical Domains Framework (TDF). The COM-B was used to create a behavioural diagnosis i.e. what needs to change; and the TDF identified relevant psychosocial drivers of the behaviour. The intervention functions, ‘education’, ‘training’ and ‘modelling’ were identified the most appropriate for changing AMR prescribing behaviour, using the COM-B mapping from the BCW. The development of the e-learning activity was underpinned by ‘gamification’ as a teaching approach, specifically using a ‘spot the difference’ format animation. There is increasing evidence of improved knowledge among healthcare professional on sepsis management [34] and antibiotic use [35], following usage of online

	games despite previous inconclusive evidence of the utility of educational games for health professionals [36].
Method: Please also elaborate how participants interacted with the illustrated consultations – perhaps providing a few examples.	We have added more details as follows in the manuscript: They were asked to engage with the e-learning activity by watching and answering questions posed within the activity. The intervention comprised two scenarios with the first depicting a consultation by a prescriber to reach a no antibiotic prescribing decision whereby patient-centred approaches [37] such as holistic care, individualised care, respectful care and empowering care, to reach a prescribing decision were absent. The overall patient experience and satisfaction was poor; the patient was not confident in the treatment decision made in the consultation. This is followed by a second scenario where a prescriber used a patient-centred motivational interviewing style [38] to reach a no antibiotic prescribing decision in the consultation. In this scenario, the patient left the consultation satisfied and confident with the treatment received. To facilitate active learning, prescribers were explicitly asked to ‘spot the difference’ between the two scenarios by answering a range of open and closed questions that were incorporated in this second scenario. These questions focused on the different approach the second prescriber undertook to consult with the patient.
Method Which were the different study components that participants had to complete and which you measured?	We explained in the manuscript that: The study was conducted in four stages in the following order: (1) pre-intervention online questionnaire, (2) e-learning intervention, (3) post-intervention online questionnaire and (4) semi-structured telephone interviews. In terms of measurement, we have added further description in the manuscript: Overall usefulness of the intervention was assessed by analysing prescribers’ self-reported confidence and knowledge in treating patients with RTIs before and after undertaking the intervention, and views on the relevance of the intervention to their work. Acceptability of the intervention was assessed in semi-structured interviews.

	The feasibility of data collection methods was assessed by recording the number of study components completed by prescribers. Responses to open and closed questions in the e-learning intervention were not collated and analysed because the aim of the study was to assess the acceptability and feasibility of implementing the intervention.
Method: Please elaborate how you compared the quantitative and qualitative data during the analysis.	We have added more detail to how both data sets were compared: Quantitative data were collected prior to the qualitative interviews and both datasets were analysed separately but within the same time frame. Quantitative findings informed the qualitative interviews and the qualitative findings explained in more detail, the quantitative findings. An iterative process of comparing both the quantitative and qualitative datasets allowed for data triangulation and confirmed the accuracy of findings in the interviews, providing further insight to the phenomenon under study [41, 45].
Discussion Could the results be further discussed in relation to the design of the intervention (both content and pedagogy) – the discussion is right now summing up the results rather than providing the reader with some insights into how to create such interventions, what are the do’s and don’ts?	We have added further discussion around the design of the intervention: Although the development of the intervention was directly informed by research conducted with prescribers with similar demographic characteristics as those in this study and underpinned by psychological and pedagogic theories, there were aspects to the intervention that requires further refinement specifically the level of difficulty of the clinical content. Intervention development, testing and evaluation is an iterative process and should be based on a user-centred approach [51, 52]. The next iteration of the e-learning intervention will focus on the addition of complex clinical content, possibly multiple activities and testing focused on the appropriateness of the clinical content.
Discussion It says the next stage is to evaluate the clinical content – how is that different from what was evaluated in this study (knowledge, suggestions	We proposed a review of the clinical content in the e-learning intervention in terms of its appropriateness. We

for improvements, etc.)? and why is it not incorporated in this study? It would also be interesting to discuss how much you can achieve by providing short, flexible, feasible interventions as compared to an intervention being a bit longer? Hence, how to balance time with learning outcome?	recognised that our phrasing was not clear and this has now been altered to: Future work will consider the addition of complex clinical content in the intervention before considering a full trial. We added further reflections an unexpected outcome reported in this study: Reflecting on one’s professional practice is a key requirement for pharmacist and nurse revalidation [49, 50] and is an interesting outcome that was not expected or measured in this study. The brief and easily accessible intervention used in this study is similar to the learning that healthcare staff are increasingly accessing e.g. ‘Espresso’ sessions held at workplaces, suggesting the potential for widespread adoption.
Discussion: Please apart from discussing generalization challenges due to a small sample size and due to participants in general being experienced, could you also reflect how the way in which you recruited participant might have influenced the results – hence, was there a selection bias which might have played a role?	We have added further limitations to our study at the end of the discussion section. A small sample was recruited to determine the acceptability and feasibility of using the intervention. Therefore, findings will need to be interpreted with caution. Prescribers were also an opportunistic sample, were generally more experienced and may therefore be biased towards appropriate prescribing. Although interviews took place as soon as possible after prescribers had completed the intervention and post-intervention questionnaire, these typically took place after 2 weeks and this time lag could affect recall of their experiences of undertaking the intervention.
Reviewer 2	
This paper describes a pilot study evaluating the feasibility acceptability to learners of a 5 minute interactive animation e-learning activity focusing on a simulated adult patient with a self-limiting upper respiratory tract infection. The learners were a highly select group consisting of 11 nurses and 4 pharmacists representing a response rate of only 7% of the prescribers contacted, at least some of whom were contacted because they had participated in previous learning experiences. The basic findings were	Thank you for your comments and reflection. We have expanded our discussion to incorporate these: Although the development of the intervention was directly informed by research conducted with prescribers with similar demographic characteristics as those in this study and underpinned by psychological and pedagogic theories, there were aspects to the intervention that requires further refinement specifically the level of difficulty of the clinical content. Intervention development, testing and evaluation is an iterative process and should be based on a user-centred approach [51, 52]. The next iteration of the e-learning intervention will focus on the addition of complex clinical content, possibly multiple activities and testing focused on the appropriateness of the clinical content.

that the the e-learning experience was feasible and that the evaluators indicated that it was acceptable.

While the activity was associated with a self-reported improved confidence in dealing with patients represented by the scenario by enforcing their preexisting knowledge and experience, the participants, 80% had been qualified for >2 years, indicated that the activity had “aimed too low”.

Thus, a major lesson of this experience is the difficulty of gauging the appropriate level of complexity in an activity such as this – something which may require iterative trials at various levels and the development of multiple e-learning activities geared to various levels of training and experience.